# Redefining Urban Recreation: A Demand Analysis for Urban Year-Round Green Exercise

**DOI:** 10.3390/ijerph21111483

**Published:** 2024-11-07

**Authors:** Konrad Reuß, Christopher Huth

**Affiliations:** Institute of Sports Science, University of the Bundeswehr Munich, 85579 Neubiberg, Germany; christopher.huth@unibw.de

**Keywords:** green exercise, urban green spaces, demand, year-round, meteorological conditions, physical activity

## Abstract

Urban green exercise offers substantial physical and mental health benefits, especially in urban environments with limited natural access. This study analyzes the demand for urban year-round green exercise under various meteorological conditions. Using a primary empirical research design, data were collected from 408 active participants in Germany through an online survey. The survey consisted of one questionnaire, with multiple sections including demographics, green exercise, year-round green exercise, and nature-relatedness. Spearman rank correlations and Pearson’s correlation were conducted for data analysis, and linear regressions calculated differences between groups. The results indicate a high demand for green exercise, with most participants engaging in endurance-based sports in urban green spaces at least once a week, regardless of weather conditions. However, adverse weather, such as precipitation and extreme cold, significantly reduces the likelihood of green exercise. Furthermore, the study identifies a positive correlation between participants’ connection to nature and their likelihood of engaging in green exercise under different weather conditions. The findings suggest that urban year-round green exercise could be a viable public health intervention, accessible to a broad demographic, with the potential to improve overall well-being. However, further research is needed to explore the barriers to green exercise in adverse weather and to develop strategies to promote year-round green exercise.

## 1. Introduction

More than half of the world’s population lives in cities, and an increase in the concentration in urban areas is expected [1,2,3]. Urbanization describes the growth of the population and the increase in densification in built-up areas [4]. The characteristics of settlement structure and transport infrastructure are among the determinants of the frequency, duration, and intensity of physical activity (PA) in the built environment [5]. An increase in PA is seen as a priority for public health and can be addressed through encouraging an active lifestyle [3]. This change can be targeted on an individual level as well through the built environment [6]. Accordingly, the design of the built environment offers the potential for promoting PA [5].

Urban green spaces as one aspect of the built environment can enable a variety of positive health and social benefits for all age groups [2]. This includes providing incentives for PA [6]. PA, which is practiced in nature, has been defined as green exercise (GE) [7,8] and is considered to provide even greater health benefits than PA in other environments [7]. However, some differences can also be attributed to different environmental conditions [9]. Interaction with nature can lead to a variety of different benefits. These range from physical and mental health to cognitive, social, and spiritual benefits [10]. Under good conditions, intensive exercise, games, and activities can be observed more frequently [5]. The increasing demand for outdoor recreation in proximity to the city increases the pressure on spaces. Therefore, suitable urban green spaces are needed within the city as spaces for GE [11].

The emergence of modern societies with advancing urbanization has, for many people, taken away the dependence on nature [10]. The psychological separation between humans and the environment is reinforced by the physical separation due to increasing urbanization and reduced opportunities to experience nature [12]. However, many outdoor activities have established themselves at a consistently high level, and GE is no longer limited to the open countryside; it is also practiced in urban areas more often [13]. Urban green spaces such as parks and green belts play a crucial role by providing infrastructure and incentives for GE [14].

The importance of GE has been emphasized in recent years, especially during the Corona pandemic [15,16]. The impact of COVID-19 resulting in various restrictions, which also concerned the practice of sports, revealed that many people have a strong desire to be outside and to be active there as well [17]. Besides the coronavirus pandemic, which caused quite some stress to the healthcare system, the German healthcare system already has the highest total health spending in the European Union [18]. GE can be an accessible and cheap form of PA, so an increase in GE should be a priority for public health measures [6]. According to the World Health Organization [19], few other public health measures can do this. However, even though GE can be cheap and accessible, it can be strongly influenced by atmospheric factors [20].

Atmospheric phenomena that contribute to the experience of nature are fresh air, sun, heat, cold, wind, or clouds [21]. Additional factors are the seasons, amount of daylight, and weather in the form of temperature, wind, or precipitation [20,22]. Bad weather, in the form of precipitation, cold, and wind, is an obstacle that leads to a strong decrease in PA [20,23]. In the winter months, slippery road conditions or the fear of slipping on ice can further reduce the amount of GE [22]. The number of total daily steps decreases significantly for the elderly in winter compared to the summer months [24]. Children are also less active during winter than at other times of the year [25]. Fewer hours of available daylight further reduce the amount of GE [26]. The decrease of GE in winter reduces the beneficial effects of PA, and the benefits of interacting with green spaces are greatly reduced as well [27]. When unpleasant weather causes people to spend more time indoors, well-being can be reduced, as the restorative effect of outdoor activities and nature is lost [28]. Rain in summer and ice or snow in winter are among the conditions that are most likely to lead to physical inactivity. Other conditions that can lead to inactivity are extreme heat and cold temperatures [22]. A lack of activity and interaction with nature reduces the benefits of GE further [27]. This reduction due to phenomena such as precipitation or cold could be counterbalanced by year-round green exercise (YRGE). For this study, urban year-round green exercise, understood as PA, is performed in urban green spaces throughout the year regardless of the atmospheric phenomena that can be encountered. To the best of the author’s knowledge, no studies specifically investigated this form of GE.

Another factor influencing GE in general can be found in the connection to nature [29]. Nature-relatedness refers to the individual’s affective, cognitive, and experiential connection with the natural world and indicates how much a person feels connected to nature and the natural elements [30]. Nature-relatedness also influences the amount of GE [31,32]; thus, nature-relatedness is seen as a moderating factor for GE [6,33].

Given the potential health benefits of GE in general and the drastic decrease of GE during winter or in times with increased precipitation, this study aims to assess the demand for urban year-round green exercise. Since there is currently no knowledge about whether there is an existing demand for urban YRGE, the research questions for this study are as follows: (1) How high is the demand for urban YRGE? (2) Are there perceived health benefits resulting from urban YRGE? (3) are the factors of socio-economic status, demographic factors, or nature-relatedness relevant for explaining engagement in urban YRGE?

## 2. Materials and Methods

The study follows a direct primary empirical research design focusing on urban YRGE. To record a holistic understanding of the demand for urban YRGE, the view of the active participants from various sports was recorded. The questionnaire was tested beforehand for clarity of wording and logical structure and revised accordingly. The internal consistency of the individual questions was checked using Cronbach’s alpha. The questionnaire was online for the period from 15 January 2024 to 15 March 2024, available in the German language. The data were collected online via SoSci Survey. The participant’s recruitment was based on the snowball principle, and no eligibility criteria were applied.

Where possible and appropriate, a five-point Likert scale with (1) disagreement/unlikely and (5) agreement/likely was used to answer the questions. These scales have been widely used since they best reflect the participant’s perspectives [34,35]. Sections from past studies were used to compile the questionnaire. The questionnaire was structured as follows (Table 1). The first section included general questions on GE [36], such as “For how much time through the course of a regular week, do you engage in activities that increase your breathing or make you sweat?” as well as questions regarding the accessibility to urban green spaces in general [37], which was assessed through questions such as “How long does it take you to get to the green space where you do your activity”. The following section recorded the self-reported GE during different meteorological conditions through eight items (α = 0.87). It included questions like “How likely are you to be active outdoors even in precipitation (rain or snow)?”. In addition, the perceived health effects of YRGE were recorded with nine items (α = 0.87), including statements such as “Regular GE makes me feel more balanced”. In the final section, the NR-6 scale (α = 0.82) was used to measure the general individual connection to nature. The NR-6 scale [38] comprises six statements, such as “I take notice of wildlife wherever I am”. The NR-6 score was calculated by averaging all six items. The higher the score, the more pronounced the connection to nature.

### Statistical Analysis

The data were initially analyzed descriptively, and a series of Spearman rank correlations and Pearson’s correlations were conducted to calculate the correlations. Differences between groups were calculated by linear regression (ANOVA) with Bonferroni post-hoc. The data were analyzed using IBM SPSS Statistics Version 29.0.2.

## 3. Results

A total of 772 questionnaires were completed for the active participants, of which 408 could ultimately be used for the evaluation. The remaining questionnaires were excluded due to missing data.

### 3.1. Participants

Of the participants, 53.95% (n = 220) are female, and 46.15% are male (n = 188). The average age of the participants is between 41 and 50 years, and the average net monthly income is between EUR 2001 and EUR 3000. Regarding educational level, 22.7% (n = 93) stated that their highest level of education was secondary school, intermediate school, or A-level. In total, 58.65% (n = 239) stated that they had a university degree (bachelor’s, master’s, doctorate, etc.), 15.75% (n = 64) stated that they had completed vocational training, and 3.05% (n = 12) stated that they were still at school or had no school-leaving qualification. In addition, 59.3% (n = 242) stated that their work is predominantly sedentary, 19.15% (n = 78) are retired, 14.75% (n = 60) do physically demanding work (e.g., haulage, construction site, etc.) or manual labor (e.g., electrical, plumbing, carpentry, etc.), and 3.95% (n = 16) have a job that is mainly outdoors (e.g., gardening, etc.). A total of 96.35% (n = 393) of the participants live in Germany.

### 3.2. Accessibility of Green Spaces and Sports Practiced

On average, the participants visit urban green spaces once a week, with 44.1% (n = 180) visiting urban green spaces two to four times a week. In general, the accessibility to the urban green spaces was rated (very) good by 94.65% (n = 386), and the participants needed on average, between 5 and 10 min to reach green spaces. A majority of 86.3% (n = 352) had access to an urban green space where they engaged in GE within a maximum of 15 min from their place of residence. To access the urban green space, 53.75% (n = 219) of the participants walked, 26.05% (n = 109) used a bicycle, 18.65% (n = 76) used a car, and 1.75% (n = 7) used public transportation. Of those respondents who visit the green spaces on foot, 98.2% (n = 215) reach these spaces within a maximum of 15 min. By bicycle, 81.15% (n = 86), by car, 64.55% (n = 49), and by public transport, 28.65% (n = 2) reach the respective spaces within this time. No differences in the time to access urban green spaces or the choice of transportation could be found for socio-demographic factors. However, a difference in the choice of transportation and the frequency of visits to urban green spaces could be found (F(3,404) = 8.730; *p* < 0.001). People who walk tend to visit urban green spaces more often than the remaining modes of transportation.

In total, GE is for 70.1% (n = 286) self-organized. The participants engaged in various individual or team sports. Most of the participants, 51.5% (n = 210), practiced endurance sports such as jogging, Nordic walking, hiking, or cycling; strength-based sports, 14.4% (n = 59,) such as calisthenics or fitness training; outdoor gymnastics, 6.9% (n = 28), such as balance training, stretching, resistance band training, or yoga; or team sports, 4.7% (n = 19), such as basketball, soccer, or volleyball. A significant difference between different sports clusters and participants’ age was found (F(6,401) = 6.587, *p* < 0.001). Younger people tend to practice team sports (µ = 3.37; SD = 1.83) and strength sports more often (µ = 3.47; SD = 1.77), and elderly people practice outdoor gymnastics (µ = 5.64; SD = 1.83) more regularly.

On average, participants engaged in moderate to vigorous physical activities (MVPA) in urban green spaces for 2–3 h per week (µ = 2.88, SD = 1.37). No significant difference in the amount of MVPA per week could be found between the different sports clusters. However, a significant difference between genders (F(5,402) = 3.95; *p* = 0.002) was found. Males (µ = 3.12; SD = 1.48) tend to be engaged in MVPA more often than females (µ = 2.67; SD = 1.22). No significant differences were found in the remaining socio-demographic factors. In general, urban GE seems to be dependent on the number of visits to green spaces as well as the mode of transportation, gender, age, and nature-relatedness (F(5,402) = 10.45. *p* < 0.001).

### 3.3. Physical Activity During Different Meteorological Conditions

About engaging in GE in (a) precipitation, 49.7% (n = 203) stated that they are (very) likely to be active during this condition, and 25.5% responded with (very) unlikely. In the case of (b) slippery or muddy ground, a total of 40.4% (n = 165) stated that they were (very) likely to be active, and 37.5% (n = 153) answered (very) unlikely. For exercising in (c) extreme cold weather, a total of 53.95% (n = 220) are (very) likely to be active, and 26.7% (n = 109) stated they were (very) unlikely to be active. For the condition (d) frozen ground or ice, 29.4% (n = 120) are (very) likely to be active outside, and 53.75% (n = 219) answered (very) unlikely. For (e) strong winds, a total of 26.95% (n = 110) are (very) likely to be active outside, and 48.05% (n = 196) answered (very) unlikely. For (f) extreme heat, a total of 50.0% (n = 204) are (very) likely to be active outside, and 23.0% (n = 94) stated it is (very) unlikely for them to be active during this condition. In (g) strong sun radiation, a total of 55.9% (n = 228) stated that they are (very) likely to be active outside, and 18.95% (n = 77) are (very) unlikely to be active. In total, 61.5% (n = 251) stated that they were (very) likely to be active outside for at least 5 min, and 15.25% (n = 62) stated that they were (very) unlikely to be active outside for at least five minutes. These findings are shown in Figure 1.

A positive significant correlation (*p* < 0.01) could be found for the frequency of visits to urban green spaces and being active during meteorological conditions, (a) (r = 0.284), (b) (r = 0.319), (c) (r = 0.303), (d) (r = 0.199), (e) (r = 0.256), (f) (r = 0.147), as well as for being active outside for at least 5 min (r = 0.209). These findings are presented in Table 2. A positive significant correlation (*p* < 0.01) could also be found in the amount of MVPA per week and being active during meteorological conditions (a) (r = 0.218), (b) (r = 0.265), (c) (r = 0.261), (d) (r = 0.218), (e) (r = 0.368), (f) (r = 0.268), (g) (r = 0.223) as well as for being active outside for at least 5 min (r = 0.266).

Significant differences between the different sports and GE during meteorological conditions could be found for (a) (F(1,401) = 3.68; *p* < 0.001), (b) (F(1,401) = 2.79; *p* = 0.011), and (c) (F(1,401) = 4.77; *p* < 0.001). The endurance sports cluster is more likely to be active during these conditions than the strength sports cluster. Furthermore, there are also differences in being active for at least 5 min (F(1,401) = 3.15; *p* = 0.005) between the endurance and strength clusters. No difference between the GE during different meteorological conditions and the socio-demographic factors could be found.

### 3.4. Nature Relatedness as a Moderating Factor

The NR-6 score has an average of 3.6 (SD = 0.8). There is a significant positive correlation between visiting green spaces and the connection to nature (r = 0.184; *p* < 0.01). Furthermore, there is also a positive significant correlation between the connection to nature and the amount of MVPA (r = 0.150; *p* = 0.002) in urban green spaces. In addition, a significant positive correlation (* *p* < 0.05; ** *p* < 0.01) was also found for the connection to nature with the probability of green exercise in meteorological conditions (a) r = 0.166 **, (b) r= 0.108 *, (c) r = 0.209 **, (d) r = 0.187 *, (e) r = 0.111 *, and (f) r = 0.109 *, as well as for an activity of at least five minutes (r = 0.146 *). These findings are presented in Table 3. No difference could be found between nature-relatedness and sports clusters or socio-demographics.

### 3.5. Perceived Benefits from Year-Round Urban Green Exercise

The perceived benefits resulting from GE are seen as a moderating factor. For most of the perceived effects, there is a positive significant correlation (* *p* < 0.05, ** *p* < 0.001) with urban GE during different meteorological conditions. Table 4 shows these findings.

Significant positive relations were also found for the self-reported health status over the last 12 months (r = 0.215; *p* < 0.001), current psychological well-being (r = 0.193; *p* < 0.001), and current social cohesion (r = 0.138; *p* = 0.005) and the frequency of urban YRGE. No difference between the types of physical activity could be found. For sociodemographic factors, no difference could be found except for age and feeling ill less often (F(7,400) = 4.84, *p* < 0.001).

## 4. Discussion

This study aimed to better understand the demand for urban YRGE. The key finding is that there seems to be a high demand for YRGE in Germany. This demand seems not to be limited by specific socio-demographic factors. However, nature-relatedness seems to play an important role in this regard. Accessibility to urban green spaces, as a primary determinant for the amount of GE, was overall rated as good. For the vast majority of participants, access to urban green spaces was possible in less than 15 min, which is within the time frame required to reach these spaces [39]. Good accessibility is seen as a determinant for the frequency of visits [33]. In this study, accessibility on foot seems to be excellent, and people who access urban green spaces by walking tend to visit these places more often. The average frequency of visiting urban green spaces one to two times per week is in line with results from previous studies [7]. According to the literature, a critical barrier for the effects of interaction with nature is five minutes [1,8]. In the present study, the majority exceeded this critical barrier during the metrological conditions commonly seen as bad weather, such as precipitation and cold temperatures. Overall, the demand for urban YRGE seems to be high. On average, the participants engaged in GE at least once a week. And almost half of the participants engaged in GE between two and four times a week. The majority of the participants engaged in endurance-based sports. These types of activities are the most frequently conducted sports in green spaces [7,40]. Due to the high number of participants who engaged in endurance sports, this result should be interpreted with caution, especially since the WHO also recommends that strength training should be performed at least twice a week [41].

The demand for strength-based urban YRGE seems to be lower. However, this might also be due to a lack of suitable sports spaces. This deficiency could be a potential barrier for urban YRGE, which could be reduced by improving existing spaces [25]^.^ Even though the general demand for urban YRGE is estimated to be high, the influence of adverse meteorological conditions, in particular the precipitation and cold temperature that occur in winter [20,23], is also reflected in this study in a decrease in the expected probability of engaging in urban GE during these conditions. The influence of heat and sunlight has less of an impact on urban GE. This study found little to no differences in urban YRGE for different socio-demographic factors, which is in line with the existing research on GE [33]. Therefore, these findings support the statement that GE is a form of PA that is accessible to all [7]. Such an accessible and cheap form of PA is especially important considering the rising healthcare costs. A strong influence for urban YRGE was found in nature-relatedness. This result indicates a similar influence of nature-relatedness to regular GE [30,32]. Since this study found little influence from socio-demographic factors but a strong influence from nature-relatedness on urban YRGE, there remains a need to understand through which factors YRGE can be promoted. Therefore, strategies to improve nature-relatedness might lead to an increase in YRGE.

Since nature connectedness is linked to spending time in nature and vice versa, more urban GE will lead to people spending more time in nature, thereby increasing their connection to nature [42], which eventually would lead to more urban YRGE. In addition, an increase in GE as well as an increase in nature-relatedness might also lead to more environmentally conscious behavior, such as active transportation, which might lead to health benefits and potentially can lead to a reduction in air pollution and greenhouse gas emissions [28] which can also lead to more health benefits. The self-reported physical and mental health effects resulting from GE also during meteorological conditions, usually perceived as bad weather, suggest that urban YRGE can lead to various health benefits. These findings align with the existing literature on the health benefits of GE in general [6,10,28]. In particular, since the items better mood, balanced, calmer, higher motivation, and higher productivity also tend to be influenced by GE during bad weather conditions, urban YRGE can improve personal well-being and health. Health effects from GE are expected to occur most frequently when people exercise during all weather conditions [29], which seems to be confirmed by the findings from this study. Perceived health benefits are seen as a moderating factor for GE [6], which could lead to people engaging in more urban YRGE. However, since those effects are only self-reported, further confirmation is needed. Nonetheless, policymakers and sports governance should try to raise awareness for the potential health benefits resulting from YRGE.

Considering the results from this study for perceived health resulting from YRGE, elderly people might benefit the most. However, only a few differences were found in this study, and urban YRGE might be considered as a possible health intervention that is accessible to all. For the vast majority, GE is conducted in a self-organized manner; therefore, future research should assess if there are sufficient offers surrounding urban YRGE that could lower the barrier for people to engage in YRGE. Therefore, clubs and associations should be encouraged to implement courses for YRGE in their program.

Due to some limitations of this study, the results should be interpreted cautiously. The main limitation of this study is its rather small sample size, as well as the fact that the participants already tend to engage regularly in YRGE. Furthermore, the composition of the sample according to socio-economic status and educational level may have a potentially significant impact on the study findings and limit the generalizability of the findings. Future studies should take these limitations into account.

Since the NR-6 scale score in this survey indicates that the participants’ connection to nature seems to be fairly strong, as well as their rather high education, this might also be seen as a limiting factor. Another limitation is that the sports practiced were predominantly endurance sports, and the probability of exercise in different weather conditions is based only on one’s own statements. The same applies to the health effects. Therefore, future studies should aim to contribute to those still-existing gaps by considering these limitations.

## 5. Conclusions

This study revealed that there is a high demand for urban YRGE. Even though the accessibility of urban green spaces in Germany is rated as good overall, the most dominant influence on the amount of YRGE conducted can be found in the individual nature relatedness. Engaging in YRGE seems beneficial for various health outcomes and is not influenced by socio-economic or demographic factors. Therefore, we conclude that urban YRGE can be a form of PA that is cheap and accessible to all. Therefore, governance and sports representatives should encourage people to engage in urban YRGE. This could be undertaken by enhancing the required sports infrastructure, such as sports spaces and facilities, and by promoting existing offers from clubs and associations for urban YRGE. Future studies should investigate if there is an adequate supply of sports infrastructure for urban YRGE.

## Figures and Tables

**Figure 1 ijerph-21-01483-f001:**
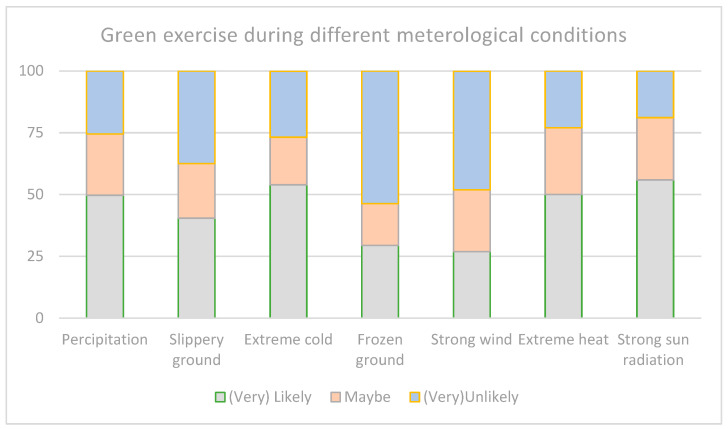
Stated probability of participants engaging in green exercise during different meteorological conditions.

**Table 1 ijerph-21-01483-t001:** Structure of the questionnaire and brief description of items.

Item	Description	Level of Scale
Demographics	Participants were asked to state their age, gender, level of education, income, profession, and residency.	Nominal and interval
Urban GE	Participants were asked to state their access to urban green spaces, mode of transportation, amount, and type of activity in green spaces,	Nominal, ordinal, and interval
Meteorological conditions and GE	Participants were asked to state the possibility of being active during different meteorological conditions.	Ordinal
Nature relatedness	Participants were asked to answer the NR-6 scale to reveal their connection to nature.	Ordinal

**Table 2 ijerph-21-01483-t002:** Spearman’s Rho correlation between the different meteorological conditions and green GE.

Variable	(a)	(b)	(c)	(d)	(e)	(f)	(g)
Frequency of visit MVPA (h/week)	0.284 ** 0.218 **	0.319 ** 0.265 **	0.303 ** 0.261**	0.199 ** 0.218 **	0.256 ** 0.368 **	0.147 ** 0.268 **	0.079 0.223 **

Note: (a) = Precipitation, (b) = slippery or muddy ground, (c) = extreme cold, (d) = frozen ground, (e) = strong wind, (f) = extreme heat, (g) = strong sun radiation. MVPA= Moderate to vigorous activity; h/week = hours per week; ** *p* < 0.01.

**Table 3 ijerph-21-01483-t003:** Spearman’s Rho correlation between nature-relatedness and GE during different meteorological conditions.

Variable	NR-6 Score	(*p*)
Precipitation	0.166 **	<0.001
Muddy ground	0.108 *	0.029
Cold (<zero degrees)	0.209 **	<0.001
Frozen ground	0.187 **	<0.001
Wind (40> km/h)	0.111 *	0.024
Heat (30> °C)	0.109 *	0.028
Sun radiation	0.093	0.060
Activity 5> min	0.146 *	0.003
MVPA (h/week)	0.150 *	0.002
Frequency of visit	0.184 **	<0.001

Note: * *p* < 0.05; ** *p* < 0.01.

**Table 4 ijerph-21-01483-t004:** Spearman’s Rho correlation between perceived benefits and GE during different meteorological conditions.

Variable	(1)	(2)	(3)	(4)	(5)	(6)	(7)	(8)	(9)
Precipitation	0.217 **	0.172 **	0.265 **	0.208 **	0.201 **	0.209 **	0.163 **	0.093	0.021
Muddy ground	0.161 **	0.113 *	0.198 **	0.179 **	0.149 **	0.169 **	0.140 **	0.055	0.012
Cold (<zero degrees)	0.221 **	0.176 **	0.256 **	0.206 **	0.223 **	0.245 **	0.162 **	0.178 **	0.009
Frozen ground	0.102 *	0.069	0.169 **	0.122 *	0.129 **	0.150 **	0.133 **	0.160 **	0.022
Wind (40> km/h)	0.097	0.060	0.155 **	0.138 **	0.188 **	0.120 *	0.074	0.098 *	0.005
Heat (30> °C)	0.089	0.058	0.109 *	0.120 *	0.136 **	0.171 **	0.140 **	0.098 *	0.030
Solar radiation	0.055	−0.009	0.076	0.102 *	0.079	0.142 **	0.110 *	0.104 *	0.026
Activity 5> min	0.158 **	0.085	0.237 **	0.257 **	0.196 **	0.190 **	0.191 **	0.125 *	0.050
MVPA (h/week)	0.151 **	0.070	0.140 **	0.165	0.228 **	0.149 **	0.131 **	0.110 **	0.032
Frequency of visit	0.134 **	0.094 *	0.153 **	0.153 **	0.138 **	0.125 **	0.130 **	0.030	0.018

Note: (1) = less often ill, (2) = less often injured, (3) = better mood, (4) = more balanced, (5) = more motivated, (6) = calmer, (7) = more productive, (8) = more creative, (9) = more spiritual. * *p* < 0.05; ** *p* < 0.01.

## Data Availability

The data supporting this study’s findings are available from the corresponding author upon reasonable request.

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
