# Peer review of "Redefining Urban Recreation: A Demand Analysis for Urban Year-Round Green Exercise"

_ijerph, 2024, doi:10.3390/ijerph21111483_

Round 1
Reviewer 1 Report
Comments and Suggestions for Authors
Dear authors;
Thank you for your efforts in conducting this study. It is a good and clearly described study. The topic is interesting, yet, in its current form, this paper cannot be considered for publication. However, I see value in the research approach and strongly encourage you to address the following points.
Abstract
· Line 11: Please provide more specific details about the survey conducted.
· Given the concise nature of the abstract, consider adding a brief overview of the data analysis just before the “Results” section.
Introduction
· The introduction effectively sets the stage for the topic. You present compelling arguments regarding urban recreation and current trends in green exercise, particularly in Germany.
· Line 31: Please include a citation following the statement about encouraging an active lifestyle.
· Line 62: This paragraph could be integrated into the previous one for better flow.
· Line 84: Please add a point (.) after “(YRGE).”
· Consider formulating study hypotheses to accompany the research question for clarity.
· Lines 88-90: The information in this section seems unnecessary and could be omitted.
Materials and Methods
· Line 94: Please clarify the name of the questionnaire used in the study.
· The overall Methods section requires more detailed definitions. Specifically, how were participants recruited? Were there any eligibility criteria for participation?
· Additionally, please add a subtitle for the statistical analyses section within the Methods.
Results
· The results and accompanying tables effectively reflect the measured variables and related findings.
· Ensure that GE is clearly represented in Figure 1 and that the related caption is descriptive.
Discussion
· The discussion is well-structured and written.
· Line 233: Consider starting the discussion with the study's aims or key findings before interpreting the results.
· While the study lays a solid foundation for understanding urban YRGE, addressing identified deficiencies could enhance its contributions to the field and inform future interventions. I believe the discussion could be strengthened from this perspective.
Conclusions
· Please present the study findings clearly, avoiding any inference at the beginning of the conclusion.
· Line 299: It would be more appropriate to move the limitations section to the end of the discussion, as its current placement disrupts the flow.
Comments on the Quality of English LanguageI believe that some minor improvements to the English in the manuscript could enhance its overall quality.
Author Response
Dear Reviewer,
We thank you for your constructive and helpful comments on the manuscript. We have tried to implement them to your satisfaction.
The changes that we conducted are highlighted in yellow within the manuscript.
Abstract:
Comment 1: Line 11: Please provide more specific details about the survey conducted.
Response 1: We added the following information: The survey consisted of one questionnaire, with multiple sections including demographics, green exercise, and year-round green exercise. This change can be found in Lines 11 - 13.
Comment 2: Given the concise nature of the abstract, consider adding a brief overview of the data analysis just before the “Results” section.
Response 2: We added the form of data analysis in the abstract. This change can be found in Lines 13-15.
Introduction:
Comment 1: The introduction effectively sets the stage for the topic. You present compelling arguments regarding urban recreation and current trends in green exercise, particularly in Germany.
Response 1: Thank you, we appreciate this comment.
Comment 2: Line 31: Please include a citation following the statement about encouraging an active lifestyle.
Response 2: The source Wolf and Robbins (2015) was added. This change can be found in Line 35.
Comment 3: Line 62: This paragraph could be integrated into the previous one for better flow.
Response 3: We added it to the previous section. It can now be found in Lines 65 and 66.
Comment 4: Line 84: Please add a point (.) after “(YRGE).”
Response 4: A point was added. This chance can be found in Line 83.
Comment 5: Consider formulating study hypotheses to accompany the research question for clarity.
Response 5: After discussing whether we should add a hypothesis we decided to reformulate the sub-questions to clarify the research question. This change can now be found in Lines 97 – 98.
Comment 6: Lines 88-90: The information in this section seems unnecessary and could be omitted.
Response 6: We deleted this information.
Material and Methods
Comment 1: Line 94: Please clarify the name of the questionnaire used in the study.
Response 1: There is no name for the questionnaire that we used. Since there is no existing questionnaire for year-round green exercise are available, we created it ourselves. Therefore, we adapted questions from previous studies (Flowers et al., 2016; Calogiuri & Elliott, 2017).
Comment 2: The overall Methods section requires more detailed definitions. Specifically, how were participants recruited? Were there any eligibility criteria for participation?
Response 2: We added the following information: The participant's recruitment was based on the snowball principle, and no eligibility criteria were applied. This change can now be found in Lines 106 & 107.
Comment 3: Additionally, please add a subtitle for the statistical analyses section within the Methods
Response 3: We added a subtitle. This change can be found in Line 127.
Results
Comment 1: The results and accompanying tables effectively reflect the measured variables and related findings.
Response 1: Thank you, we appreciate this comment.
Comment 2: Ensure that GE is clearly represented in Figure 1 and that the related caption is descriptive.
Response 2: We implemented this comment in the figure and changed the description. This can now be found in Lines 199 & 200.
Discussion
Comment 1: The discussion is well-structured and written.
Response 1: Thank you, we appreciate this comment.
Comment 2: Consider starting the discussion with the study's aims or key findings before interpreting the results.
Response 2: The key finding was added at the start of the discussion. This can now be found in Lines 246 – 249.
Comment 3: While the study lays a solid foundation for understanding urban YRGE, addressing identified deficiencies could enhance its contributions to the field and inform future interventions. I believe the discussion could be strengthened from this perspective.
Response 3: We added some information for future contributions. These can be found in Lines 268 & 269, 300-302, and 38 & 309.
Conclusion
Comment 1: Please present the study findings clearly, avoiding any inference at the beginning of the conclusion.
Response 1: We implemented this comment by changing the structure of the conclusion. This can now be found in Lines 324-334.
Comment 2: Line 299: It would be more appropriate to move the limitations section to the end of the discussion, as its current placement disrupts the flow.
Response 2: We moved the limitations at the end of the discussion. These can now be found in Lines 312-322.
Comments on the Quality of English Language: I believe that some minor improvements to the English in the manuscript could enhance its overall quality.
Response on the Quality of English Language: We checked the entire manuscript and tried to reduce spelling and grammatical errors.
We hope that we understood your comments correctly and changed them accordingly.
Thank you very much for your support and your contributions to our research.
Kind regards
Konrad Reuß
Reviewer 2 Report
Comments and Suggestions for Authors
Dear Authors,
First of all, I congratulate you on your study.
Below you can find some suggestions that I think will improve the quality of your article.
The research makes a significant contribution to understanding the demand for urban green exercise and its potential health benefits. Clearly defining practical recommendations and strategies will enhance the practical value of the study.
It may be beneficial to provide a clearer definition of “urban year-round green exercise” in the title.
In the introduction, more explicit identification of gaps in the literature could strengthen the necessity of the study.
To enhance the methodological rigor of the research, more information should be provided regarding the validity and reliability of the scales used. Additionally, the criteria for participant selection should be explained.
The socio-economic status and education level of the sample may limit the generalizability of the findings. It is particularly important to emphasize this limitation in the discussion section. Furthermore, more examples of how the proposed strategies can be practically implemented should be included in the discussion.
In the conclusions, it is important to emphasize the contributions of urban green exercise to public health. Providing concrete recommendations for practitioners and policymakers in this section would be more beneficial.
Comments on the Quality of English LanguageIn general, the English of the article can be improved at a minor level so that the research can be understood more clearly by the interested readers.
Author Response
Dear Reviewer,
We thank you for your constructive and helpful comments on the manuscript. We have tried to implement them to your satisfaction.
The changes that we conducted are highlighted in yellow within the manuscript
Comment 1: It may be beneficial to provide a clearer definition of “urban year-round green exercise”.
Response 1: We added a clearer definition. This can now be found in Lines 81-85.
Comment 2: In the introduction, more explicit identification of gaps in the literature could strengthen the necessity of the study.
Response 2: We edited the introduction to more explicitly mention the research gap. These changes can be found in Lines 85 and 86.
Comment 3: To enhance the methodological rigor of the research, more information should be provided regarding the validity and reliability of the scales used. Additionally, the criteria for participant selection should be explained.
Response 3: We added more information on the participant criteria, which can now be found in Lines 106 & 107, as well as information on the questionnaire's validity, which can now be found in Lines 110 & 111.
Comment 4: The socio-economic status and education level of the sample may limit the generalizability of the findings. It is particularly important to emphasize this limitation in the discussion section. Furthermore, more examples of how the proposed strategies can be practically implemented should be included in the discussion.
Response 4: We added the socio-economic status and educational level as limiting factors in the discussion. This change can now be found in Lines 312 – 314. We also added more strategies to the discussion. These changes can now be found in Lines 282 & 283, 300-302, and 318 & 309.
Comment 5: In the conclusions, it is important to emphasize the contributions of urban green exercise to public health. Providing concrete recommendations for practitioners and policymakers in this section would be more beneficial.
Response 5: We specified the practical implementation. This change can now be found in Lines 329-334.
Comments on the Quality of English Language: In general, the English of the article can be improved at a minor level so that the research can be understood more clearly by the interested readers.
Response on the Quality of English Language: We checked the entire manuscript and tried to reduce spelling and grammatical errors.
We hope that we understood your comments correctly and changed them accordingly.
Thank you very much for your support and your contributions to our research.
Kind regards
Konrad Reuß
Reviewer 3 Report
Comments and Suggestions for Authors
The manuscript selects an important topic in public health, year-round urban green exercise (YRGE). The research design is sound, and the data collection methods and statistical analyses are scientifically robust. However, there is room for improvement in the articulation of the research objectives, presentation of results, and depth of discussion. Adding more specific data reports, enhancing comparisons with existing literature in the results discussion, and further refining the language will significantly improve the scientific value and readability of the manuscript.
Introduction
The manuscript clearly states the research objectives, which are to assess the demand for YRGE and explore its moderating factors. However, the rationale for selecting these research objectives and their scientific significance needs to be strengthened to help readers better understand the importance of the study. It is recommended to provide more background on the choice of research objectives, particularly how these objectives address gaps in the existing literature.
Methods
The study employs a direct empirical research design with a sample size of 408 participants. The questionnaire structure and scale design are appropriate, and the statistical methods include descriptive and correlation analyses. However, detailed parameters for the internal consistency test are missing. It is suggested to include specific Cronbach's alpha values in the data analysis section to enhance transparency and credibility. Additionally, some important statistical results (e.g., correlation coefficients and significance levels) are not presented in tables, which limits readers' understanding of the findings. Including detailed tables to present the results would help readers grasp the analytical conclusions more intuitively.
Results
The results clearly present participants' responses to GE under different meteorological conditions. However, the figures and tables used should correspond to the report in the results section. For instance, Tables 2, 3, and 4 are not mentioned in the report. Moreover, abbreviations in figures should be explained in the captions.
Discussion
The discussion on comparing results with existing literature is limited. It is recommended to provide more detailed comparisons between your findings and existing research, particularly on the impact of adverse weather conditions on physical activity. Additionally, more relevant studies can be cited to deepen the discussion. Some sentences need further refinement to ensure clarity and flow. The manuscript mentions some methodological limitations, such as the small sample size and participants' preference for YRGE, which is good. However, further discussion on the potential impact of these limitations on the study's findings and how future research could address them would be beneficial. Furthermore, suggestions for future research could include specific strategies to improve interventions, such as how to increase YRGE participation under adverse weather conditions.
Other Comments
The overall language of the article is good, but some parts could be made smoother in terms of grammar and logical structure. For example, the sentence: "Even though the general demand for urban YRGE is estimated to be high the influence of meteorological conditions, in particular the precipitation and cold temperature that occur in winter, is also reflected in this study in a decrease in the expected probability of engaging in urban GE during these conditions." It is suggested to conduct a thorough language review to reduce spelling and grammatical errors, thus improving the overall quality of the manuscript.
Comments on the Quality of English Language
Need to do some grammar check
Author Response
Dear Reviewer,
We thank you for your constructive and helpful comments on the manuscript. We have tried to implement them to your satisfaction.
The changes that we conducted are highlighted in yellow within the manuscript.
Comment 1 (Introduction): It is recommended to provide more background on the choice of research objectives, particularly how these objectives address gaps in the existing literature.
Response 1: We added a section at the end of the introduction to specify why the research objective was chosen. This change can now be found in Lines 92 – 95.
Comment 2 (Methods): However, detailed parameters for the internal consistency test are missing. It is suggested to include specific Cronbach's alpha values in the data analysis section to enhance transparency and credibility. Additionally, some important statistical results (e.g., correlation coefficients and significance levels) are not presented in tables, which limits readers' understanding of the findings. Including detailed tables to present the results would help readers grasp the analytical conclusions more intuitively.
Response 2: We implemented the Cronbach´s alpha. This can be found in Lines 103 & 104, 118, 120, and 121. Further, we also give the significant level in the notes below the Tables. This can now be found in Lines 211, 239, 231 and 239.
Comment 3 (Results): However, the figures and tables used should correspond to the report in the results section. For instance, Tables 2, 3, and 4 are not mentioned in the report. Moreover, abbreviations in figures should be explained in the captions.
Response 3: We mentioned the tables in tables in the report. These changes can now be found in Lines 204, 227, and 235. Further, we also changed the abbreviation. The change can be found in lines 199 & 200.
Comment 4 (Discussion): It is recommended to provide more detailed comparisons between your findings and existing research, particularly on the impact of adverse weather conditions on physical activity. Additionally, more relevant studies can be cited to deepen the discussion.
Response 4: We implemented this comment. Changes can now be found in Lines 253, 272, 275, 280, and 282 & 283.
Comment 5 (Discussion): The manuscript mentions some methodological limitations, such as the small sample size and participants' preference for YRGE, which is good. However, further discussion on the potential impact of these limitations on the study's findings and how future research could address them would be beneficial.
Response 5: We implemented this comment in Lines 316 – 318.
Comment 6 (Discussion): Furthermore, suggestions for future research could include specific strategies to improve interventions, such as how to increase YRGE participation under adverse weather conditions.
Response 6: We implemented this comment in Lines 329-334.
Comments on the Quality of English Language: It is suggested to conduct a thorough language review to reduce spelling and grammatical errors, thus improving the overall quality of the manuscript. Need to do some grammar check.
Response on the Quality of English Language: We checked the entire manuscript and tried to reduce spelling and grammatical errors.
We hope that we understood your comments correctly and changed them accordingly.
Thank you very much for your support and your contributions to our research.
Kind regards
Konrad Reuß
Round 2
Reviewer 3 Report
Comments and Suggestions for Authors
The authors have made thoughtful and targeted revisions in response to the previous review comments. Each suggested change has been addressed comprehensively, and the manuscript has been strengthened as a result. The clarity of the research objectives, the robustness of the methodology, and the coherence of the findings have all been improved. The authors’ responses to specific critiques were detailed and effective, demonstrating a clear effort to align the manuscript with the standards of the journal.
In my opinion, the revised manuscript has now reached an acceptable level of quality, addressing all major concerns satisfactorily. Therefore, I recommend this manuscript for acceptance.